# Immunohistochemical Expression of Epithelial Cell Adhesion Molecule (EpCAM) in Salivary Gland Cancer: Correlation with the Biological Behavior

**DOI:** 10.3390/diagnostics13162652

**Published:** 2023-08-11

**Authors:** Ioanna Kalaitsidou, Nikoleta Pasteli, Gregory Venetis, Athanasios Poulopoulos, Konstantinos Antoniades

**Affiliations:** 1Department of Oral and Maxillofacial Surgery, Aristotle University of Thessaloniki, 54124 Thessaloniki, Greece; gvenetis@dent.auth.gr (G.V.);; 2Department of Cranio-Maxillofacial Surgery, Inselspital, Bern University Hospital, University of Bern, CH-3010 Bern, Switzerland; 3Pathology Department, G. Papanikolaou Hospital, 57010 Thessaloniki, Greece; 4Department of Oral Medicine and Maxillofacial Pathology, Aristotle University, 54124 Thessaloniki, Greece; akpoul@dent.auth.gr

**Keywords:** salivary gland malignancy, perineural invasion, EpCAM, prognosis, biological behavior

## Abstract

Salivary gland neoplasms comprise a diverse group of tumors with different biological behaviors and clinical outcomes. Understanding the underlying molecular alterations associated with these malignancies is critical for accurate diagnosis, prognosis, and treatment strategies. Among the many biomarkers under investigation, epithelial cell adhesion molecule (EpCAM) has emerged as a promising candidate in salivary gland cancer research. This article aims to provide a comprehensive overview of the differential expression of EpCAM in salivary gland cancer and its potential correlation with the biological behavior of these tumors. The clinical characteristics of 65 patients with salivary gland malignancy of different histopathological subtypes were included. We report the differential expression of EpCAM and the relationship between the clinical and histopathologic features of these tumors. Regarding the evaluation of the effect of EpCAM expression on survival, in our study, we showed that tumors with high EpCAM expression had reduced disease-free survival (DFS) and overall survival (OS) (*p* < 0.001) compared to patients with cancers with low EpCAM expression. In addition, the concurrent presence of perineural invasion and positive EpCAM expression appeared to be associated with shorter disease-free survival and overall survival. In conclusion, our study confirmed the prognostic value of detecting perineural invasion and EpCAM expression.

## 1. Introduction

Salivary gland neoplasms comprise a diverse group of tumors with different biological behaviors and clinical outcomes. The incidence of malignant neoplasms of the salivary glands varies among different researchers, with an estimated incidence of 0.4–2.6:100,000. They account for 3–6% of head and neck malignancies and less than 1% of all malignancies [1,2,3]. Despite their rarity, salivary gland neoplasms exhibit a diversity that is arguably unmatched by any other organ [4]. There are more than 20 different types of malignant salivary gland tumors [5]. This diversity in histology may contribute to the variable clinical behavior and prognosis of salivary gland malignancies [6]. Understanding the underlying molecular alterations associated with these malignancies is critical for accurate diagnosis, prognosis, and treatment strategies. Among the many biomarkers under investigation, epithelial cell adhesion molecule (EpCAM) has emerged as a promising candidate in salivary gland cancer research [7].

EpCAM is a transmembrane glycoprotein that plays a critical role in cell adhesion and signaling. It is widely expressed in normal epithelial tissues and has been found to be dysregulated in several types of cancer, including salivary gland cancer [8]. Aberrant EpCAM expression has been associated with tumor initiation, progression, metastasis, and resistance to therapy in several malignancies. Therefore, studying the role of EpCAM in malignant salivary gland neoplasms and its correlation with the biological behavior of these tumors may provide valuable insights into their pathogenesis [9].

The purpose of this research is to study the expression patterns of EpCAM in malignant salivary gland neoplasms and to explore its potential association with tumor characteristics, clinical outcomes, and therapeutic response. We explore the differential expression of EpCAM in different histologic subtypes of salivary gland malignancies, including adenoid cystic carcinoma, mucoepidermoid carcinoma, acinic cell carcinoma, and others. In addition, we investigate the potential correlation between EpCAM expression levels and clinicopathological parameters such as tumor grade, perineural invasion and lymphatic infiltration, distant metastasis, and patient survival.

## 2. Materials and Methods

### 2.1. Tissue Samples

The study included patients with malignant tumors of the salivary glands of different histopathologic subtypes who were treated at the Department of Oral and Maxillofacial Surgery of “G. Papanikolaou” General Hospital of Thessaloniki from 1 January 2007 to 31 December 2016. Available information from individual medical records was recorded, including demographic (age, sex) and clinical data of all cases, and the histopathologic diagnosis was confirmed by a review re-evaluation of hematoxylineosin slides, using WHO 2017 criteria for tumor classification of the salivary glands [5]. All methods and experimental protocols using human tissues (formalin-fixed, paraffin-embedded (FFPE) tissues) were performed according to the relevant guidelines and regulations approved by the Institutional Review Board of the “G. Papanikolaou” General Hospital of Thessaloniki, Aristotle University of Thessaloniki (1137/24-08-2016). Informed consent was waived because the IRB determined that this retrospective study presented a minimal risk to patients (risk level I).

Of the total 93 cases of salivary gland cancer, a sufficient number of cases (28) were excluded from the study because they did not meet the requirements of the research protocol. The reasons for exclusion were: -Missing data from the patient’s medical record;-Short postoperative follow-up (<5 years);-Tumor type (squamous cell carcinoma, lymphoma, secondary malignancies);-Poor conditions and amounts of neoplastic tissue; and-Quality of immunohistochemical staining.

### 2.2. Immunohistochemical Methods

Immunohistochemical staining of all cases was performed using the fully automated Leica BOND-MAX™ immunohistochemistry system (Leica Biosystems Newcastle Ltd., Newcastle Upon Tyne, UK) with the complete bond two-step, polymer, free biotin, and short-chain immunohistochemistry probe system Bond Polymer Refine Detection (Leica Biosystems Newcastle Ltd.) and EpCAM/epithelial-specific antigen (MOC-31) mouse monoclonal antibody (Cell Marque, Sigma-Aldrich Co. LLC., Rocklin, CA, USA). Samples of colonic adenocarcinoma were used as positive controls and sections omitting the primary antibody were used as negative controls.

In this study, we used a semi-quantitative approach to evaluate the expression of EpCAM. The evaluation was based on three factors: the intensity score (IS), the product of the intensity score and the proportion score (PS), and the total immunostaining score (TIS). The IS represents the intensity of staining relative to control cells, graded from 0 (no staining) to 3 (strong staining). Meanwhile, the PS represents the proportion of positively stained tumor cells and is scored as 0 (none), 1 (<10%), 2 (10–50%), 3 (51–80%), or 4 (>80%). The TIS is the result of multiplying the IS by the PS, resulting in values ranging from 0 to 12. However, there are only nine possible TIS values (0, 1, 2, 3, 4, 6, 8, 9, and 12). Using the TIS, we categorized the samples into two groups: the low-expression group (TIS 0–8) and the high-expression group (TIS 9 and 12). To investigate the cellular distribution of EpCAM, we analyzed its expression pattern and determined whether it was predominantly membranous or cytoplasmic. 

Two independent pathologists (KM and PN) reviewed the results of immunohistochemical staining microscopically. The interobserver variability of the grading results was low (<3%). In cases where there was disagreement between the pathologists, the staining was re-evaluated using a multiview microscope and the cases were discussed until agreement was reached.

### 2.3. Statistical Analysis

Statistical analysis was performed using SPSS software (IBM^®^ SPSS^®^ Statistics), V.27.0, https://www.ibm.com/spss (accessed on 6 May 2023). Different tests were used depending on the specific purpose: Student’s *t*-test, Fisher’s exact test, and Pearson’s chi-squared test were used to analyze various clinicopathologic parameters. The Kaplan–Meier method and the log-rank test were used to evaluate patient survival statistics. Univariate and multivariate analyses were performed using the Cox regression model to evaluate factors influencing patient survival. A significance level of 5% was used, and results with a *p*-value ≤ 0.05 were considered statistically significant.

## 3. Results

The clinical characteristics of sixty-five patients with salivary gland malignancy of different histopathological subtypes were included in our study. The salivary gland neoplasms studied were adenoid cystic carcinoma with 18 cases (27.7%), mucoepidermoid carcinoma with 20 cases (30.8%), polymorphous adenocarcinoma with 9 cases (13.8%), adenocarcinoma (NOS) with 4 cases (6.2%), salivary duct carcinoma with 5 cases (7.7%), epithelial–myoepithelial carcinoma with 5 cases (7.7%), acinic cell carcinoma with 2 cases (3.1%), and carcinoma-ex pleomorphic adenoma with 2 cases (3.1%). Concerning the gender among the patients, 35 were female (53.8%) and 30 were male (46.2%), with a female-to-male ratio of 1:1.16. The mean age was 58.7 years (range 16–89 years). Regarding the site, 30 (46.15%) cases involved the major salivary glands, and 35 (53.85%) cases involved the minor salivary glands (Table 1).

### 3.1. EpCAM Expression

Overall, out of 65 malignant neoplasms of the salivary glands, the evaluation regarding the expression of EpCAM was positive in 81.5%, with diversity regarding the proportion score (PS), intensity score (IS), and total immunostaining score (TIS). Immunohistochemical expression of EpCAM was both cytoplasmic and nuclear (Figure 1).

The correlation between EpCAM expression by IS, TIS, and clinicopathological parameters is shown in Appendix A.

Regarding the intensity score (IS), a significant correlation with the type of carcinoma was observed (*p*-value < 0.001), and more specifically, a higher percentage of patients with adenocystic carcinoma (66.7%) and patients with mucoepidermoid carcinoma (40.0%) had moderate staining, whereas patients with adenocarcinoma and polymorphous adenocarcinoma had a higher percentage of weak staining (75% and 66.7%, respectively). Finally, all patients with salivary duct carcinoma had intense staining.

A significant correlation was also observed with the site in the major salivary glands (*p*-value = 0.027), where 45.8% of patients with parotid gland tumors also had intense staining, whereas patients with submandibular and sublingual salivary gland neoplasms had a higher percentage (75% and 100%, respectively) of moderate staining.

There is a further correlation between the intensity score (IS) and recurrence (local) and distant metastasis. In carcinomas with local recurrence, intense staining was observed in 83.3%, while in those without local recurrence, the largest percentage, 40.7%, had moderate staining (*p*-value = 0.006). In cases with distant metastases, 50% had moderate staining, while 41.5% of patients without distant metastases had weak staining (*p*-value < 0.001).

Finally, a significant correlation was observed between perineural invasion and vascular infiltration. In carcinomas with perineural invasion, moderate and intense staining was observed in 44.8% and 44.8%, respectively, whereas in specimens without perineural infiltration, weak staining was observed in 47.2% (*p*-value < 0.001). In specimens with vascular infiltration, moderate staining was observed in 48.1%, while in specimens without vascular infiltration, weak staining was observed in 42.1% (*p*-value < 0.001).

Regarding positively stained tumor cells, a significant correlation with the type of carcinoma was observed (*p*-value < 0.001). Specifically, the highest percentage of adenocystic carcinoma cases (50%) had 51–80% of positively stained tumor cells. In contrast, a percentage of positively stained tumor cells less than 10% was observed in cases with adenocarcinoma, polymorphous adenocarcinoma, and epithelial–myoepithelial carcinoma at rates of 75%, 66.7%, and 60%, respectively. Finally, all patients with salivary duct carcinoma had >80% positively stained tumor cells.

There is an additional correlation between positively stained tumor cells and local disease recurrence and distant metastasis. In carcinomas with local recurrence, 83.3% had a percentage of positively stained tumor cells >80%, while in cases without local recurrence, the largest percentage, 30.5%, had positively stained tumor cells <10% (*p*-value = 0.004). A total of 45.8% of patients with distant metastases had positive tumor cells above 80%, while 43.9% of patients without distant metastases had positive tumor cells below 10% (*p*-value < 0.001).

Finally, a significant correlation was observed between perineural invasion and vascular and muscle infiltration. The percentage of positive cells greater than 80% was found in 44.8% of carcinomas with perineural invasion, while the percentage of positive cells less than 10% was found in 50% of cases without perineural invasion (*p*-value < 0.001). Regarding vascular infiltration, a percentage of positive cells >80% was found in 44.4% of cases. The percentage of positive cells <10% was found in 44.7% of cases without vascular infiltration (*p*-value < 0.001). Muscle infiltration had a percentage of positive cells of 10–50% in 31.4% of cases, while cases without muscle infiltration had a percentage of <10% in 40.0% of cases (*p*-value = 0.013).

Regarding the total immunostaining score, a significant correlation with the type of carcinoma was observed (*p*-value = 0.001), and more specifically, the largest percentage of cases with adenocystic carcinoma (38.9%) had moderate expression (total immunostaining score 6.8). On the other hand, cases with adenocarcinoma, mucoepidermoid carcinoma, polymorphous adenocarcinoma, and epithelial–myoepithelial carcinoma had a higher percentage of low expression (TIS 1.2), 100%, 55%, 66.7%, and 60%, respectively. Finally, all cases (100%) with salivary duct carcinoma had strong expression (TIS 9.12). 

There is also a correlation between local recurrence and distant metastasis. In cases with local recurrence, 83.3% had strong expression, while in cases without local recurrence, the majority, 52.5%, had weak expression (*p*-value = 0.003). A total of 45.8% of cases with distant recurrence had strong expression, while 68.3% of cases without distant recurrence had weak expression (*p*-value < 0.001).

Finally, a significant correlation was observed between perineural invasion and vascular infiltration. In carcinomas where perineural invasion was observed, strong expression was present in 44.8%, while without perineural invasion, weak expression was present in 72.2% (*p*-value < 0.001). In specimens with vascular infiltration, strong expression was observed in 44.4%, while without vascular infiltration, weak expression was observed in 57.9% (*p*-value = 0.002).

### 3.2. Survival Analysis

During follow-up, 33 of 65 (50.8%) patients experienced tumor recurrence, and 21 (32.3%) patients died. In tumors with high EpCAM expression, 18 of 24 patients relapsed, and 17 of 24 patients died. Kaplan–Meier analysis showed that patients with EpCAM-positive tumors had reduced disease-free survival (DFS) (*p* < 0.001) and overall survival (OS) (*p* < 0.001) compared to patients with low EpCAM-positive tumors. In tumors with perineural invasion (PNI), 22 of 29 patients relapsed, and 18 of 29 died. Kaplan–Meier analysis showed that patients with tumors with perineural invasion had reduced disease-free survival (DFS) (*p* < 0.001) and overall survival (OS) (*p* < 0.001) compared to patients with cancers without perineural invasion. In tumors with concurrent perineural invasion and high-grade EpCAM expression (PNI+/EpCAM+), 17 of 21 cases relapsed, and 16 of 21 died. Kaplan–Meier analysis showed that patients with PNI+/EpCAM+ expressing tumors had statistically significantly reduced overall survival (OS) (*p* < 0.001) and disease-free survival (RFS) (*p* < 0.001) compared to patients with PNI−/EpCAM− expressing tumors. However, due to the small number of patients with two positive markers, this specific finding needs to be verified in a larger group of patients (Figure 2).

## 4. Discussion

Malignant salivary gland tumors are usually present in the sixth and seventh decades of life and have a male predominance. However, age at diagnosis and gender prevalence vary according to histologic type [10,11]. In our study, regarding the gender of the patients, 35 were women (53.8%) and 30 were men (46.2%), and the mean age was 58.7 years.

It is generally accepted that salivary gland cancer is inherently highly heterogeneous due to its long course and wide histopathologic diversity, resulting in many subcases and clinical manifestations that are not accurately predicted by its management guidelines. On the one hand, the unique epidemiological data of salivary gland cancer result in studies that focus on its therapeutic approach and include relatively small numbers of patients compared to other cancer types. As a result, the guidelines cannot accurately and reliably predict and cover all possible forms and stages of the disease. This is because it is difficult to collect a sufficient number of patients to complete the necessary amount of information on all the possible histologic types, which, as developed in the general part, exceed 20, their stages, degree of differentiation, and combinations. Thus, in the treatment of these patients, therapeutic dilemmas arise, such as the choice or not of neck dissection, radiotherapy, or more radical surgical removal with the possible sacrifice of the facial nerve in parotid tumors, due to the lack of sufficient relevant data from the literature [10].

The relative deficiencies in treatment protocols are complemented and fed back by the lack of sufficient documentation of prognostic factors outside the clinics. Characteristically, the objective difficulties in data collection, aggregation, and processing have resulted in an even smaller body of literature on the possible prognostic factors of the disease relative to the epidemiology. Known prognostic factors for the progression of salivary gland cancer include stage, histologic type, degree of differentiation, perineural invasion, and a number of clinical factors such as initial presenting symptoms, pain, skin infiltration, facial nerve injury, and others. Often there is no complete agreement or confirmation of these factors in the studies conducted [3,12]. Spiro et al. [12], in 1973, in a series of 492 patients, showed the primary value of clinical stage in predicting the clinical course of patients with salivary gland cancer. The first comprehensive study with the sole objective of finding and identifying possible prognostic factors was performed in 1986 by Tran et al. [13], who identified sex, age, side of development, clinical stage, histologic type, and status of surgical margins as possible influencing variables. From the statistical analysis of the 113 patients, clinical stage and differentiation were found to be influential factors. Andersen et al. [14] in 1991, in a series of 95 patients with a follow-up of 25 years, found that in submandibular, sublingual, and minor salivary gland cancers, the most common histologic type was adenoid cystic carcinoma. In this study, the type of salivary gland involved, the stage of the disease, and the location of the tumor were identified as important prognostic factors. On the contrary, Anderson et al. [15] in their 1995 study focused exclusively on small salivary gland carcinomas and identified as positive prognostic factors the classification into stage I and II, the absence of cervical metastases, and free surgical margins. In the study by McHugh et al. [16] (2012) in a series of 115 patients, the 5-year survival rate was 79% and the negative prognostic factors identified were the presence of cervical lymph node metastases, extracapsular lymph node infiltration, and perineural invasion.

From the list of studies dealing with the prognosis of salivary gland cancer, it is clear that there is agreement on a number of factors, such as stage, cervical lymph node infiltration, and the presence of perineural invasion in the specimens. These factors are considered given and guaranteed and were calculated as such in our study. Our study showed that the incidence of perineural invasion was 42.62%, which is consistent with the trend of the overall incidence of the presence of perineural invasion in malignant neoplasms of malignant glands, ranging from 23% to 96%. In terms of location, the majority of tumors with perineural invasion involved the major salivary glands, primarily the parotid gland. These results are consistent with other studies using heterogeneous anatomic sites [17,18]. The tumors studied in our study are adenoid cystic carcinoma with 18 cases (27.7%), mucoepidermoid carcinoma with 20 cases (30.8%), polymorphous adenocarcinoma with 9 cases (13.8%), adenocarcinoma (NOS) with 4 cases (6.2%), salivary duct carcinoma with 5 cases (7.7%), epithelial–myoepithelial carcinoma with 5 cases (7.7%), acinic cell carcinoma with 2 cases (3.1%), and carcinoma–ex pleomorphic adenoma with 2 cases (3.1%). The highest incidence of perineural infiltration was reported in salivary duct carcinoma (5/5) and adenoid cystic carcinoma (13/18). These findings are consistent with other studies [19,20,21].

The clinical and histologic features of perineural invasion have been extensively studied in various types of cancer, and it is well-known that perineural invasion is often clinically silent [22]. However, in this study, perineural infiltration was significantly associated with facial nerve dysfunction in patients with malignant parotid gland cancer at the time of diagnosis, as facial nerve palsy was present in 71.4% of parotid gland cancer patients whose specimens showed perineural invasion. Similar to our findings, Huyett et al. [20] in 2018 showed that facial nerve palsy was present in the majority of patients with parotid malignancies with positive PNI at the time of presentation. In the present study, the presence of PNI was only significantly associated with lymphovascular infiltration. Our results are in agreement with a number of published studies [20,23].

Evaluation of the effect of perineural infiltration on patient survival is controversial. The role of perineural infiltration as a prognostic factor in salivary gland carcinoma is controversial. In several studies, perineural invasion was not a predictor of worse survival [20,21,24,25,26]. Interestingly, in our study, we found that perineural infiltration was strongly associated with 5-year overall survival and disease-free survival in patients with salivary gland carcinoma, despite the small sample size and different pathologies of the salivary glands. This is consistent with previous reports showing that PNI was significantly associated with worse survival, suggesting that PNI has prognostic significance in malignant salivary gland carcinoma [26,27,28,29,30,31,32].

In recent years, the role of immunohistochemistry has had a catalytic effect on the therapeutic management of cancer patients in general. The ability to detect the expression of specific antigens in the cells of various neoplasms has allowed the development of monoclonal antibodies, which have made it possible to target specific antigens and have opened very wide and promising horizons in the treatment of cancer in general. In the field of salivary gland cancer, there is currently no immunohistochemical factor that can be considered of absolute diagnostic or prognostic value. The role of a number of molecules in the development and progression of neoplasms has been studied with varying results, but without any of them being widely used to play a decisive role in the diagnostic or therapeutic process [33,34].

Epithelial cell adhesion molecule (EpCAM) expression can be detected in various human epithelial tissues, such as the glandular or ductal epithelium of the gastrointestinal tract, respiratory tract, kidney, gallbladder, and salivary glands. Its expression level varies in different tissues. Normal stratified squamous epithelial cells do not express EpCAM. It is noted that epithelial cells with high proliferative activity tend to overexpress EpCAM, while reduced EpCAM expression is often detected in differentiated cells. As a result, most epithelial neoplasms show increased EpCAM expression. Its expression may also be related to the stages of neoplastic growth [9].

For diagnostic purposes, differential EpCAM expression, alone or in combination with other markers, can help differentiate several neoplasms with overlapping histopathologic features. For example, basal cell carcinomas of the skin with squamous metaplasia have higher EpCAM expression than basaloid squamous cell carcinomas. Hepatocellular carcinomas can be distinguished from metastatic adenocarcinomas or cholangiocarcinomas by their general lack of EpCAM expression. Similarly, lung adenocarcinomas often overexpress EpCAM, whereas mesotheliomas are consistently negative [35,36,37].

The prognostic relevance of EpCAM expression has also been demonstrated in various types of cancer. Several studies found that increased EpCAM expression was associated with advanced stage and poor survival in patients with breast cancer, gallbladder cancer, ovarian cancer, Vater tubercle and esophageal cancer, and oral squamous cell carcinoma, because it acts as an inhibitor of E-cadherin. Therefore, EpCAM is thought to play an important role in local recurrence and distant metastasis. However, studies in gastric cancer, clear renal cell carcinoma, colorectal cancer, and non-small cell lung cancer have reported a direct correlation between increased EpCAM expression and an overall better prognosis of patients [35,36,37,38,39,40].

Few studies in the literature have investigated the expression of EpCAM in salivary gland neoplasms and the correlation of histopathologic findings with the biological behavior of these tumors. Therefore, the aim of this study was to investigate the expression of EpCAM in malignant salivary gland neoplasms and its relationship with their biological behavior.

In this study, we report the differential expression of EpCAM in eight types of malignant salivary gland neoplasms and the relationship between the clinical and histopathologic features of these tumors and the degree of expression. EPCAM expression in salivary glands is generally predominantly membranous. This means that EPCAM is mainly located on the cell membrane of salivary gland epithelial cells. Its presence in the membrane allows it to interact with other molecules and participate in cell adhesion and signaling events. However, some reports have suggested that the cytoplasmic distribution of EpCAM varies according to tumor type and histologic differentiation of the carcinoma. Therefore, it may have diagnostic value [37,38] In our study, we detected both membranous and cytoplasmic expression in the cancer cells of the neoplasms studied.

Phattarataratip et al. [41] studied the expression of epithelial cell adhesion molecules in various salivary gland neoplasms; including mucoepidermal carcinoma, adenoid cystic carcinoma, pleomorphic adenoma, and polymorphous adenocarcinoma. They included EpCAM and showed different expression patterns of EpCAM among salivary gland neoplasms. Similarly, in our study, we observed different expression patterns of EpCAM among the studied carcinomas.

They showed that reduced EpCAM expression was associated with aggressive features in mucoepidermoid carcinoma, whereas adenocystic carcinoma (AdCC) showed negative or weakly positive EpCAM immunoreactivity. In contrast, a statistically significant association between strong EpCAM expression and tumor aggressiveness was observed in both mucoepidermoid and adenocystic carcinomas in our study. This is consistent with the results of the study by Lee et al. [42]. However, both the previous studies and ours had some limitations because the number of cases included in the study was very small.

A previous study investigated the expression of tumor-associated calcium signal transducer 2 (TACSTD2, Trop2), a homolog of EpCAM, in salivary adenocystic carcinoma (ACC) [42]. Overexpression of TACSTD2 was associated with poor prognosis in patients, although the molecule does not reflect the histologic type. Several investigators have raised the potential prognostic significance of EpCAM overexpression in various cancers [43,44,45]. Similarly, EpCAM overexpression was associated with high histologic differentiation and distant metastasis. For example, in our study, in mucoepidermoid carcinoma, all cases with poor differentiation (100%) showed strong expression of EpCAM (*p* < 0.001). In contrast, no statistically significant difference was observed between EpCAM expression and the degree of differentiation, as 50.0% of patients with adenocystic carcinoma had well-differentiated and strong staining, patients with intermediate differentiation had both moderate staining (75.0%) and strong staining (25.0%), and poorly differentiated patients (75.0%) had moderate staining (*p*-value = 0.104).

Regarding the evaluation of the effect of EpCAM expression on survival, in our study, we showed that tumors with high EpCAM expression had reduced disease-free survival (RFS) (*p* < 0.001) and overall survival (OS) (*p* < 0.001) compared to patients with cancers with low EpCAM expression. The study by Lee et al. also confirmed that high EpCAM expression is an independent factor influencing patient survival. Of course, in this study, the authors only examined the expression of EpCAM in adenoid cystic carcinoma [42].

Finally, it should be noted that in our study, a statistically significant reduction in overall survival (OS) (*p* < 0.001) and disease-free survival (DFS) (*p* < 0.001) was observed in tumors with both perineural invasion and high levels of EpCAM expression (PNI+/EpCam+) compared to patients with PNI−/EpCam− cancers. In addition, the concurrent presence of perineural invasion and positive EpCAM expression appeared to be associated with shorter disease-free survival and overall survival. However, due to the small number of patients with two positive markers, this specific finding needs to be verified in a larger group of patients.

## 5. Conclusions

The expression of EpCAM in salivary gland cancer shows variability, with some cases demonstrating overexpression while others exhibit lower levels. This phenomenon can be attributed to several factors. A plausible hypothesis to explain this variation includes tumor heterogeneity and genetic mutations: Salivary gland cancer is known for its heterogeneous nature, with tumors differing significantly in their genetic makeup and cellular characteristics. Different subtypes of salivary gland cancer can have different molecular profiles, resulting in different levels of EpCAM expression. Tumors with elevated EpCAM expression may belong to a specific subtype that is more dependent on this protein for cell growth and survival. Additionally, disease progression and metastasis may influence EpCAM expression. As cancer advances or spreads to other parts of the body (metastasize), tumors may acquire more aggressive traits. Consequently, they could express higher levels of EpCAM as part of their adaptive survival and invasion strategies.

In conclusion, the clinical prognostic factors of salivary gland cancer (histologic type, differentiation) are well established in the literature, while the evidence for a number of biological factors is mixed. Our study confirmed the prognostic value of detecting perineural invasion and EpCAM expression. Further studies with more samples will help to document the potential utility of EpCAM as an immunohistochemical marker in the study of prognosis in salivary gland malignancies. 

## Figures and Tables

**Figure 1 diagnostics-13-02652-f001:**
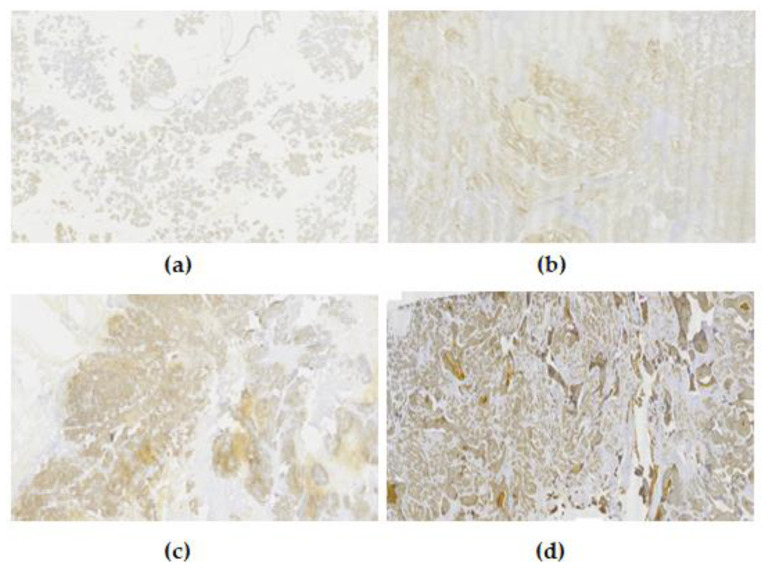
Examples of the intensity levels of EpCAM. (**a**) Negative, 0; (**b**) weak, 1+; (**c**) moderate, 2+; (**d**) strong, 3+.

**Figure 2 diagnostics-13-02652-f002:**
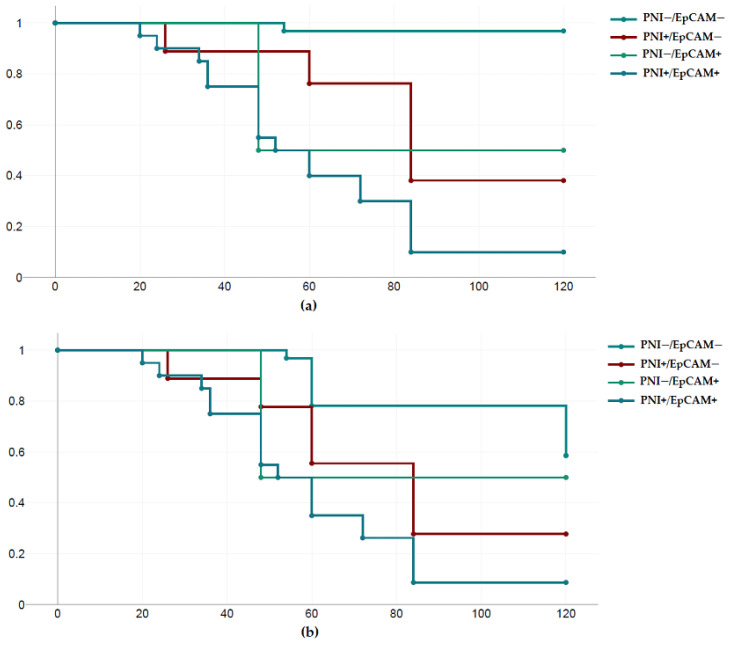
Survival analysis according to the EpCAM expression and perineural invasion. (**a**) Overall survival; (**b**) disease-free survival.

**Table 1 diagnostics-13-02652-t001:** Clinical features of 65 patients with salivary gland cancer.

Histopathological Subtypes of Salivary Gland Cancer	*n*
Adenocarcinoma (NOS)	4
Adenoid cystic carcinoma (AdCC)	18
Mucoepidermoid carcinoma (MEC)	20
Polymorphous adenocarcinoma (PAC)	9
Epithelial–myoepithelial carcinoma	5
Acinic cell carcinoma (AcCC)	2
Salivary duct carcinoma	5
Carcinoma–ex pleomorphic adenoma	2
Age	
≤50	21
>50	44
Gender	
Male	30
Female	35
Site	
Major salivary glands	
Parotid gland	24
Submandibular gland	4
Sublingual gland	2
Minor salivary glands	
Palate	21
Tongue/floor of mouth	4
Upper lip	2
Lower lip	2
Retromolar mucosa	3
Buccal mucosa	3

## Data Availability

The data presented in this study are available on request from the corresponding author.

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
