# Peer review of "Immunohistochemical Expression of Epithelial Cell Adhesion Molecule (EpCAM) in Salivary Gland Cancer: Correlation with the Biological Behavior"

_diagnostics, 2023, doi:10.3390/diagnostics13162652_

Round 1
Reviewer 1 Report
Manuscript is well written and discusses the expression of EpCAM in salivary gland malignancies. However, it could be improved further.
1. The immunohistochemical expression (IS, PS and TIS) have been mentioned in the results section of the text. A table or chart showing the scores for various parameters like tumor type, presence/absence of perineural invasion, distant metastasis, etc. would be more helpful.
2. Authors mention that the expression of EpCAM was higher in some cases, and lower in others. They have not discussed about the possible reasons for this variation. A small note, or hypothesis, as to why EpCAM is overexpressed in some cases and not in others may be useful.
3. Discussion is too lengthy. The initial paragraphs of the discussion that describe about the epidemiology and statistics of salivary gland malignancies can be made more concise. These paragraphs distract the focus of the manuscript, which is about EpCAM expression and biological behavior.
Reviewer 2 Report
It would be better to:
- add statistical analysis methods to abstract.
- move lines 57-63 from introduction to discussion section.
- remove lines 64-66 in order to prevent repeated statement.
- add informed consent to M&M section.
- add study registration code to M&M section.
- reform lines 364-371 and 358-363 as an one paragraph.
- add some explanations about possible confounding variable and limitations of this cross sectional study.
Reviewer 3 Report
Review on Kalaitsidou et al’s study entitled Immunohistochemical expression of epithelial cell adhesion molecule (EpCAM) in salivary gland cancer. Correlation with biological behaviour.
1. The abstract contains the most important findings.
2. Introduction: there are 3 main purposes of the study, but the paragraph between lines 57-63 fits to the conclusions of the study. Nothing to do with introduction.
3. It is a pity that 28 patients must have been excluded. as salivary gland tumours are rare. In such a case (rarities) I would have used even partial data to increase the number of cases. In this situation the number of records and available data is also published in the study. I know that it is harder to work with.
4. Tables: table 1. do not split into 2 pages. Pearson correlation data (R value and p value) can be displayed in a cross table like display. In this way it is easier to understand relationships between factors, and you may even flag significant correlations.
5. PNI as an abbreviation pops up out of nowhere (line 297). Please, define it!
6. So conclusively: EpCAM is an independent factor that has a good prognostic value in terms of both survival rates. There could be different expression levels in different salivary tumours (mucoepidermoid and adenocystic cc) in which even reduced levels of expression could be a bad prognostic factor.
No significant problems with English . Between lines 364-384 the content is described in a bit complicated manner. If possible please shorten and thereby simplify your explanation.
Reviewer 4 Report
The authors of this study were aimed to provide a comprehensive overview of the differential expression of EpCAM in malignant salivary gland neoplasms and its potential correlation with the biological behavior of these tumors.
The issue is important, moreover they conduct comprehensive results and discussion. Therefore, I have no further comments against the manuscript.
